



# Simulating net ecosystem exchange under seasonal snow cover at an Arctic tundra site

Victoria R. Dutch[1], Nick Rutter[1], Leanne Wake[1], Oliver Sonnentag[2], Gabriel Hould Gosselin[2], Melody Sandells[1], Chris Derksen[3], Branden Walker[4], Gesa Meyer[5], Richard Essery[6], Richard Kelly[7], Phillip Marsh[4], Julia Boike[8,9], Matteo Detto[10]

[1] Department of Geography and Environmental Sciences, Northumbria University, Newcastle upon Tyne, UK
[4] Département de géographie, Université de Montréal, Canada
[3] Climate Research Division, Environment and Climate Change Canada, Toronto, Canada
[4] Cold Regions Research Centre, Wilfrid Laurier University, Waterloo, Canada
[5] Climate Research Division, Environment and Climate Change Canada, Victoria, Canada
[6] School of Geosciences, University of Edinburgh, UK
[7] Department of Geography and Environmental Management, University of Waterloo, Canada
[8] Alfred Wegener Institute, Helmholtz Centre for Polar and Marine Research, Potsdam, Germany
[9] Geography Department, Humboldt-Universität zu Berlin, Germany
[10] Department of Ecology and Evolutionary Biology, Princeton University, USA

*Correspondence to*: Victoria Dutch (victoria.dutch@polarnetwork.org) for modelling and Oliver Sonnentag (oliver.sonnentag@umontreal.ca) for eddy covariance data.

## Abstract

Estimates of winter (snow-covered non-growing season) $CO_2$ fluxes across the Arctic region vary by a factor of three and a half, with considerable variation between measured and simulated fluxes. Measurements of snow properties, soil temperatures and net ecosystem exchange (NEE) at Trail Valley Creek, NWT, Canada, allowed evaluation of simulated winter NEE in a tundra environment with the Community Land Model (CLM5.0). Default CLM5.0 parameterisations did not adequately simulate winter NEE in this tundra environment, with near-zero NEE ($< 0.01$ g C m$^{-2}$ d$^{-1}$) simulated between November and mid-May. In contrast, measured NEE was broadly positive (indicating net $CO_2$ release) from snow cover onset until late April. Changes to the parameterisation of snow thermal conductivity, required to correct for a cold soil temperature bias, reduced the duration for which no NEE was simulated. Parameter sensitivity analysis revealed the critical role of the minimum soil moisture threshold on decomposition ($\Psi_{min}$) in regulating winter soil respiration. The default value of this parameter ($\Psi_{min}$) was too high, preventing simulation of soil respiration for the vast majority of the snow-covered season. In addition, the default rate of change of soil respiration with temperature (Q10) was too low, further contributing to poor model performance during winter. As $\Psi_{min}$ and Q10 had opposing effects on the magnitude of simulated winter soil respiration, larger negative values of $\Psi_{min}$ and larger positive values of Q10 are required to simulate wintertime NEE more adequately.





## 1.0: Introduction

Although considerably more attention has been paid to Arctic $CO_2$ fluxes during the growing season, winter (i.e. snow-covered non-growing season) $CO_2$ emissions are now understood to make a significant contribution to annual carbon budgets in Arctic environments (e.g. Campbell, 2019; Natali et al., 2019; Rafat et al., 2021). The cumulative effect of winter emissions may even offset plant uptake of $CO_2$ in the growing season, particularly as the climate warms (Belshe et al., 2013; Christiansen et al., 2012; Jeong et al., 2018), with the magnitude of non-growing season emissions likely to increase under climate change

(Box et al., 2019; Commane et al., 2017; Watts et al., 2021). However, understanding of non-growing season $CO_2$ fluxes is limited (Lüers et al., 2014). $CO_2$ fluxes across the Arctic region quantified by either Terrestrial Biosphere Models (TBMs) or empirical estimates vary by a factor of three and a half (377 – 1301 Tg Carbon; Natali et al., 2019).

Uncertainties in process representation and parameterisation of TBM simulations of carbon fluxes limit our ability to assess and predict future changes (Braghiere et al., 2023; Treharne et al., 2022), particularly shifts in the timing and duration of the

non-growing season. The representation of biogeochemical cycles in TBMs is subject to a high degree of parametric uncertainty (Fisher et al., 2019), with non-growing season processes and mechanisms poorly represented (Larson et al., 2021). Model intercomparison studies show large differences between individual predictions, with uncertainty in many aspects of the Arctic carbon cycle greater than the absolute magnitude of carbon fluxes (Fisher et al., 2014). Variability in carbon flux estimates between models are particularly prevalent during the winter (Fisher et al., 2014) and fluxes in the early winter

shoulder season are likely underestimated (Commane et al., 2017). Improving (or even just including) the representation and influence of snow, soil and biogeochemical non-growing season processes in TBMs will potentially improve our understanding of carbon dynamics and projections of Arctic climate change (Campbell and Laudon, 2019).

Mechanisms of non-growing season soil respiration, particularly the impact of environmental controls on heterotrophic respiration in subfreezing soils, are poorly represented in models leading to large uncertainties (Tao et al., 2021). Poor

simulation of early winter respiration in many TBMs is possibly linked to underestimation of soil temperature (Commane et al., 2017) because below-ground thermal processes become disconnected from the above-ground energy balance due to the insulation provided by nascent snow cover. This problem continues to impact soils throughout the entire snow-covered period. Such cold biases in wintertime soil temperature can be mitigated with a change in the parameterisation of snow thermal conductivity (Dutch et al., 2022; Royer et al., 2021) because the stratigraphic and hence insulative properties of Arctic

snowpacks are not well simulated (Barrere et al., 2017; Domine et al., 2019). Decreasing snow thermal conductivity, which increases near surface soil temperatures, has been found to increase simulated non-growing season net ecosystem exchange (NEE), with winter emissions more than doubling in the TBM LPJ-GUESS after addition of a multi-layer snow scheme with temporally evolving snow properties (Pongracz et al., 2021). Biases in NEE, where simulated NEE is lower than measured NEE, has previously been noted in CLM5.0 in Arctic environments (Birch et al., 2021) and other Earth System Models (Wieder

et al., 2019), due to model underestimates of $CO_2$ uptake by Arctic vegetation (Rogers et al., 2017). While this is particularly pertinent to growing season simulations, this can also impact the 'shoulder seasons' of snow cover onset and snowmelt within



the non-growing season as CLM5.0 has limited skill in reproducing the timing of key phenological events, such as leaf onset and senescence (Birch et al., 2021). Additionally, the empirical formulae used by many TBMs to model relationships between soil temperature, moisture, and soil respiration are often derived from datasets which under-sample or do not include high-latitude regions (Bonan, 2019). For example, the temperature sensitivity of soil respiration is typically described with the use of a single, globally averaged Q10 value, representing the proportional change in respiration with a 10°C rise in soil temperature (Lloyd and Taylor, 1994). However, Q10 is likely temperature-dependent (Hamdi et al., 2013; Lloyd and Taylor, 1994; Kirschbaum, 1995) and may also be influenced by other environmental conditions such as soil moisture, texture and plant community composition (Chen et al., 2020; Curiel Yuste et al., 2004; Meyer et al., 2018), although Mahecha et al. (2010) suggests otherwise. As a result, observed Q10 from studies of Arctic ecosystems are typically larger than globally averaged values, with the synthesis of Chen et al. (2020) finding a median Q10 for tundra ecosystems (5.4) approximately double that of their global median (2.3). However, differences between Arctic and global Q10 values are not reflected in Arctic climate simulations, with approximately half of the 11 models investigated by Huntzinger et al. (2020) using Q10 only half the size of observed values. Empirical relationships between soil moisture and respiration (often parameterised using soil water potential, Ψ) in many TBMs are derived from small scale studies which do not account for respiration from frozen soils (Andrén and Paustian, 1987; Orchard and Cook, 1983). Relationships between soil moisture and respiration are also likely to be influenced by other soil properties, such as bulk density, texture and carbon content, with different relationships observed for mineral and organic soils (Moyano et al., 2012). Interactions between temperature, moisture and respiration suggest that these properties should be considered together when working to improve our understanding of $CO_2$ fluxes.

As much of the Arctic tundra is snow-covered for up to 10 months of the year (Olsson et al., 2003), it is important to accurately simulate non-growing season carbon emissions under snow-covered conditions to better quantify annual carbon budgets. In our previous study (Dutch et al., 2022) examining the parameterisation of snow thermal conductivity in the Community Land Model version 5 (CLM5.0) at Trail Valley Creek (TVC), NWT, we found a cold soil temperature bias of ~ 6 °C, and suggested this bias may impact the simulation of NEE during the snow-covered non-growing season. TVC makes an ideal type-site for much of the Arctic tundra, having been intensively studied and used to characterise the hydrology of tundra regions since the mid-1990s (e.g. Marsh et al., 2008; Pomeroy et al., 1993; Quinton and Marsh, 1999). In this study, we assess whether the default parameterisation of CLM5.0 accurately simulates carbon fluxes (NEE) during the snow-covered non-growing season at TVC. We evaluate the impact on the simulation of NEE of the parameterisation of:

1)  snow thermal conductivity ($K_{eff}$),

2)  the relationship between soil moisture and soil decomposition ($\Psi_{min}$),

3)  the rate of change of soil respiration as a function of soil temperature (Q10).

The overall aim is to compare simulations of soil respiration and NEE to eddy covariance (EC; Baldocchi, 2003) measurements for 3 snow-covered non-growing seasons and consider how to parameterise the model better in Arctic tundra environments on both sub-seasonal timescales and cumulatively throughout the snow-covered non-growing season.





## 2.0: Methods

### 2.1: Study site and data

Model evaluation was undertaken with data from at Trail Valley Creek (68°45'N, 133°30'W), a mineral upland tundra site in the Inuvialuit Settlement Region, northeast of Inuvik, NWT, Canada. Mean annual air temperature at TVC was -7.9 ℃ for the period 1999 – 2018 (Grünberg et al., 2020), with typical maximum snow depths of < 50 cm (King et al., 2018). Precipitation was measured using a Geonor T-200B weighing gauge with an Alter-style wind screen and corrected as per Pan et al. (2016), as gauge under-catch is common in these types of environments (Smith, 2008; Watson et al., 2008; Gray and Male, 1981). Daily precipitation totals were disaggregated to hourly timesteps, based on the fraction of daily precipitation at each hourly timestep from ERA5 reanalysis data (Hersbach et al., 2020). Air temperature and relative humidity were measured at 2 m using a temperature/humidity sensor (Vaisala HMP35CF, Vaisala Oyj, Helsinki, Finland). Shortwave and longwave radiation were measured at a height of 4.08 m using Kipp and Zonen CNR1 and CNR4 net radiometers (Kipp & Zonen, Delft, The Netherlands). Wind speed and direction were measured at 6.1 m using an R.M. Young 05103-10 Wind Monitor (R.M. Young, Traverse City, Michigan). Discontinuous radiation measurements between January 2013 and December 2019 were gap-filled following Essery et al. (2016); gaps of 4 hours or less were filled by linear interpolation whereas longer gaps used ERA5 reanalysis data (Hersbach et al., 2020).

Measurements of NEE from the TVC EC tower (Helbig et al., 2016; Martin et al., 2022) were compared with model simulations. Measured half-hourly $CO_2$ fluxes were calculated from wind speeds measured by a Campbell Scientific CSAT3 sonic anemometer and $CO_2$ concentrations measured by an EC150 open-path $CO_2/H_2O$ infrared gas analyser at a frequency of 10 Hz at a height of 4.08 m above the ground. Net $CO_2$ fluxes are presented as NEE; we follow the micro-meteorological convention where release to the atmosphere is positive NEE and net uptake of $CO_2$ by the land surface is negative NEE.

Non-growing season NEE measurements are presented as a comparison data set to simulated NEE, primarily to assess the direction (positive or negative) of $CO_2$ fluxes and broad seasonal trends, rather than absolute magnitudes. A cautious interpretation of measured NEE is prudent due to the difficulties in operation of open-path infrared gas analysers in Arctic winter climates (Amiro, 2010; Goulden et al., 2006; Jentzsch et al., 2021a; Jentzsch et al., 2021b), frequent power failures common to meteorological stations in remote areas without line power, and low signal to noise ratios in post-processing flux corrections. Processing of EC measurements followed the pipeline described in Helbig et al. (2017):

1) remove spikes in high frequency timeseries (Vickers and Mahrt, 1997),

2) correct sonic temperatures for humidity effects (Van Dijk et al., 2004),

3) correct sonic anemometer tilt using a double rotation,

4) calculate half hourly fluxes (EddyPro v6.0+, Li-COR Biosciences),

5) apply the Webb-Pearman-Leuning (WPL)

6) fill gaps in the NEE time series where possible (Reichstein et al., 2005).



Data quality, identified using a QWPL flag (Jentzsch et al., 2021a), and availability (gap-filled and non-gap-filled) are presented in Section 1 of the Supplementary Material. Herein, the final processed and gap-filled NEE half-hourly time series are presented as weekly averages throughout the non-growing season with uncertainties calculated as standard deviations of residuals from the gap-filling algorithm (Lasslop et al., 2008).

## 2.2: Model Description

CLM5.0 (Lawrence et al., 2019) is a community-developed land surface model, which includes biogeophysics, the carbon cycle and vegetation dynamics as a TBM, within the overall framework of the Community Earth System Model (CESM; Danabasoglu et al., 2020). CLM5.0 can be run at a range of spatial scales, from a 1D point to grid cells across the entire earth surface. Recent developments relevant to modelling Arctic biogeochemical cycling include new representations of snow and soil hydrology and changes to carbon allocation schemes (Lawrence et al., 2019).

CLM5.0 describes tundra environments using a C3 Arctic Grass Plant Functional Type (PFT) (Schädel et al., 2018), with land cover data generated at a 0.5° resolution (Lawrence and Chase, 2007). However, for 1D simulations at TVC we prescribed land cover distribution as 60% C3 Arctic Grass, 33% Broadleaf Deciduous Boreal Shrub PFTs, and 7% bare ground in line with ground-based species counts within the TVC EC footprint (Voigt, Pers. Comm.).

CLM5.0 uses a vertically resolved CENTURY-type soil decomposition scheme as outlined in Koven et al. (2013). Cryoturbation, the mixing of soil material due to freeze thaw processes, was switched on for these simulations and model spin-up. The maximum depth for cryoturbation was set to 1 m, in line with observations of active layer thickness at this site (Wilcox et al., 2019). The parameterisation of soil freezing in CLM is given in Yang et al. (2018). For each layer ($j$) of the 20 biogeochemically active soil layers (the upper 8.5 m of the soil column), carbon moves through 3 soil pools with different default turnover times. The default turnover time ($K_0$) of each of these pools is modified by the rate of decomposition:

$$K_j = K_{0,j} r_T r_W r_O r_Z \qquad (1)$$

where $r_T$, $r_W$, $r_O$ and $r_Z$ are rate modifiers applied to each pool in each layer, which scale the rate of decomposition ($K_j$) depending on the soil layer temperature, soil moisture, oxygen content and depth, respectively. In this study, we focus on the soil decomposition rate modifiers $r_T$ (temperature) and $r_W$, (moisture), which are explained in more detail below.

The influence of temperature on decomposition is parameterised using a Q10 function for both frozen and unfrozen soils:

$$r_T = Q_{10}^{\left(\frac{T_j - T_{ref}}{10}\right)} \qquad (2)$$

where Q10 defines the temperature sensitivity of soil respiration, $T_j$ equals the temperature of soil layer $j$, and $T_{ref}$ is a reference temperature with a default value of 25℃. By default, CLM5.0 uses a globally constant Q10 of 1.5 (Foereid et al., 2014) for both frozen and unfrozen soils (Lawrence et al., 2018).





The scalar for the impact of soil moisture on decomposition takes the form described by Andrén and Paustian (1987):

$$r_W = \sum_{j=1}^{5} \begin{cases} 0 & \text{for } \Psi_j < \Psi_{min} \\ \dfrac{\log(\Psi_{min}/\Psi_j)}{\log(\Psi_{min}/\Psi_{max})} w_{soil,j} & \text{for } \Psi_{min} < \Psi_j < \Psi_{max} \\ 1 & \text{for } \Psi_j > \Psi_{max} \end{cases} \tag{3}$$

where $\Psi_j$ is the soil water potential in soil layer $j$, and $\Psi_{min}$ and $\Psi_{max}$ are the upper and lower limits, default values of -2 MPa
and -0.002 MPa respectively, for soil water potential to impact the rate of soil decomposition. When $\Psi$ is a greater absolute
value than $\Psi_{max}$, a change in the moisture content of the soil has no impact on rates of carbon turnover. When $\Psi$ is smaller than
$\Psi_{min}$, the soil moisture is too low for decomposition to be simulated. This is noted to be a major limitation on the respiration
from frozen soils by Lawrence et al. (2018). Respiration of previously decomposed carbon may still occur when $\Psi$ is less than
$\Psi_{min}$, up until the point where labile carbon stocks are depleted (Lawrence et al., 2018).

Parameterisation of effective snow thermal conductivity ($K_{eff}$) in CLM5.0 is after Jordan (1991). A quadratic equation is used
to infer the relationship between the density of the snow (calculated from the masses of ice and interstitial air) and the thermal
conductivity of the snowpack. Other parameterisations of this relationship, typically using different constants in the same
quadratic equation (e.g. Sturm et al., 1997; Calonne et al., 2019; Yen, 1981), were expanded upon for CLM5.0 in Dutch et al.
(2022). These different constants have been calculated from snow samples from different environments, with the
parameterisation of Sturm et al. (1997) derived from snowpacks in the Alaskan Arctic.

To simulate 1D processes at the TVC EC tower, CLM5.0 was run in point mode adjusting two gridded land surface
parameterisations as per Dutch et al. (2022). In order to better represent 1D processes, the snow accumulation factor, a scaling
factor which determines the likeliness of a sub-gridcell area to become snow-covered after a snowfall event, was increased
(Swenson and Lawrence, 2012) from 0.1 to 2.0 which is more representative of the binary nature of snow presence or absence
at a point. Additionally, the standard deviation of elevation set to 0.5 m after Malle et al. (2021). Soil sand, silt and clay
fractions (28% sand, 36% silt, and 36% clay) were taken from the mineral soil texture data set (Bonan et al., 2002), and
CLM5.0 default soil organic matter fractions (Hugelius et al., 2013) were also used.

## 2.3: Experiment Setup

The sensitivity of simulated NEE was evaluated in comparison with measured NEE in response to changes in the model
parameterisation of 1) snow thermal conductivity, 2) the relationship between soil moisture and soil decomposition ($r_w$;
Equation 3), and 3) the relationship between soil respiration and soil temperature ($r_T$; Equation 2). Simulation sensitivity was
evaluated over snow cover dates simulated by CLM5.0 (9 Oct 2016 – 23 May 2017; 12 Oct 2017 – 30 May 2018; 24 September
2018 – 23 May 2019), which were always within a week of observed snow cover onset and melt-out (Dutch et al, 2022). We
compared two options for the parameterisation of effective snow thermal conductivity, that of Jordan (1991) as used by default





in CLM5.0, and that of Sturm et al. (1997) which has been shown to improve soil temperature simulation in both CLM5.0 (Dutch et al., 2022) and other land surface models (Royer et al., 2021). Such an improvement likely occurs as the parameterisation of Sturm et al. (1997) was derived from Arctic snowpack measurements, whereas that of Jordan (1991) was based on the laboratory experiment of Yen (1962) which used sieved snow with a denser and more homogenous structure than observed in Arctic snowpacks.

We also adjusted the soil decomposition rate modifiers ($r_T$ and $r_W$ in Equation 1), similar to the approach of Tao et al. (2021), sampling a broad range of values for the parameters $\Psi_{min}$ (for $r_w$; Equation 3) and Q10 (for $r_t$; Equation 2), as listed in Table 1. Values for Q10 sampled a wide range of measured Q10 from Arctic soils (based on Chen et al., 2020; Elberling, 2007; Elberling and Brandt, 2003; Grogan and Jonasson, 2005; Mikan et al., 2002; Schmidt et al., 2008), and values of $\Psi_{min}$ were based on Tao et al. (2021). We note that the most negative values (< -200 MPa) of $\Psi_{min}$ used by Tao et al. (2021) and herein are unlikely

to be physically representative (Liang et al., 2022).

In total, 32 model simulations were performed, perturbing $K_{eff}$, Q10 and $\Psi_{min}$ simultaneously. Simultaneous perturbation of parameters avoided the one-at-a-time approach typical of many sensitivity analyses in order to examine the interaction between parameters (Gao et al., 2020) and evaluate their relative importance in improving wintertime carbon flux simulations. Simulations were spun-up for 512 years, using 128 concatenated loops of 4 years (2013 to 2016) of meteorological forcing

data. Spin up was achieved once all 3 soil carbon pools in the decomposition scheme were in a steady state. Steady state was achieved when mean annual changes in the size of the pools were less than 10 g C m$^{-3}$ for the last 10 years of the simulation and when the size of the soil carbon pools was within the range of observed values for the Mackenzie Delta region given in Figure 1 of Schuur et al. (2015). CLM5.0 simulations were run for the period 2013-2019, but only evaluated from the onset of snow cover in 2016 due to the availability of coincident snow, soil and eddy covariance measurements.

**3.0: Results**

**3.1: Measured NEE and soil temperatures**

Measured NEE was broadly positive (with weekly NEE averages ranging from -0.1 to 1.1 g C m$^{-2}$ day$^{-1}$) throughout the snow-covered non-growing season, suggesting that $CO_2$ was emitted from the ground at TVC throughout the winter (Fig. 1a; Supp. Fig. 3). Measured mean NEE was positive until mid-April, at which point measured NEE followed an increasingly negative

trend, indicating potential photosynthetic uptake. Soil freeze-up began with the onset of snowfall in October, with weekly mean 10 cm soil temperatures reaching a minimum value of -10.2 ℃ in early March. Soils began to warm as the snowpack melted, with observed weekly mean soil temperatures becoming positive in the second week of June (Fig. 1c). As considerably more NEE measurements were available for the snow-covered period of 2017 – 2018 than 2016 – 17 or 18 – 19 (Supp. Table 1), we primarily focused on 2017 – 2018 when presenting measurements or comparing measured and simulated fluxes.

However, cumulative simulated fluxes were presented for all three winters.



### 3.2: Simulated NEE

The default parameter configuration of CLM5.0 simulated negligible, near-zero NEE (all values below 0.01 g C m$^{-2}$ d$^{-1}$) between late November and mid-May in all 3 winters. CLM5.0 does not simulate Gross Primary Productivity (GPP) during the entirety of the snow-covered season in all 3 winters. Autotrophic respiration is similarly negligible (all values below 0.01

g C m$^{-2}$ d$^{-1}$) in all simulations of the snow-covered non-growing season, regardless of parameter choices. As heterotrophic respiration, other than soil biota, are also not simulated during periods of snow cover, simulated NEE and soil respiration can be considered equivalent for simulations of snow-covered non-growing seasons.

Sensitivity analysis of three parameters ($\Psi_{min}$, Q10 and snow thermal conductivity) resulted in considerable variability in the simulated soil respiration and NEE over all three snow-covered periods (Fig. 1; Supp. Fig. 3). Minimum total snow-covered

non-growing season NEE was simulated for the default $\Psi_{min}$ (-2 MPa) and the default Jordan (1991) snow thermal conductivity parameterisation. For all years, simulated fluxes were greatest for a Q10 of 1.5, $\Psi_{min}$ of -2000 MPa, and the Sturm et al. (1997) snow thermal conductivity parameterisation. Simulated cumulative NEE spanned 370 g C m$^{-2}$ in magnitude between the different sets of parameter values (Fig. 2). This difference in cumulative simulated NEE was greater in years with earlier snow onset date, e.g. 2018-19, as this increased the duration of relatively warmer winter soils with higher respiration rates during

freeze-up, in comparison with the total duration of colder soils throughout the non-growing season snow cover. In all 3 winters, simulations were most sensitive to chosen parameter values during the freeze up period, with the range of soil respiration fluxes approximately double that in midwinter (Fig. 1). Simulated NEE decreased gradually from snow cover onset until December-January, and then remained at that level until late April when NEE increased as soils warm and snow melts.

Changes from Jordan (1991) to Sturm et al. (1997) representations of snow thermal conductivity delayed, by approximately 2

months, the onset of moisture limitation for simulations with the default value of $\Psi_{min}$, enabling more positive simulation of NEE during freeze-up. The choice of snow thermal conductivity scheme significantly impacted simulations of mean winter soil respiration when considered throughout the total snow-covered non-growing season in all three years (Student's t-test: $t_{16-17}$ = -6.76, $t_{17-18}$ = -8.01, $t_{18-19}$ = -8.02, p < 0.001). Compared to the default Jordan (1991) parameterisation of snow thermal conductivity, the Sturm et al. (1997) parameterisation resulted in warmer near surface soil (Fig. 1c) and hence more positive

NEE, provided soil respiration had not become moisture limited. Model sensitivity to $\Psi_{min}$ was lower for the Jordan (1991) snow thermal conductivity parameterisation (Fig. 3a) than for Sturm et al. (1997) (Fig. 3b); differences between parameterisations were greatest with a more negative $\Psi_{min}$ (Fig. 3c).

Simulated winter soil moisture potentials ($\Psi_j$; Eq. 2) had a typical value of approximately -15 MPa, lower than the default $\Psi_{min}$ (-2 MPa), preventing soil decomposition and respiration for the majority of the winter. Analysis of variance showed significant

differences between simulated mean snow season soil respiration ($F_{16-17}$ = 19.45, $F_{17-18}$ = 22.41, $F_{18-19}$ = 23.80, p < 0.001) and cumulative snow season NEE ($F_{16-17}$ = 19.47, $F_{17-18}$ = 22.45, $F_{18-19}$ = 23.86, p < 0.001; Fig. 2) for $\Psi_{min}$ of -2 and -2000 MPa, though differences between simulations with only one order of magnitude between their $\Psi_{min}$ were not always deemed





statistically significant ($\alpha$ = 0.001). Consequently, adjusting $\Psi_{min}$ had the largest impact on simulated fluxes, with larger negative $\Psi_{min}$ resulting in larger NEE.

Changes to Q10 had a smaller impact on simulated NEE than the parameterisation of $K_{eff}$ or $\Psi_{min}$, with analysis of variance showing no significant difference between the mean snow season soil respiration for different Q10 (Table 1) in all 3 winters. Differences in simulated cumulative snow season fluxes were also not statistically significant. Additionally, simulation sensitivity to frozen Q10 values (Schmidt et al., 2008) were tested. An extreme frozen Q10 of 300, after Schmidt et al. (2008), did not reduce the gap between simulated and measured NEE, with no appreciable difference between model runs where all other parameter choices were held constant.

Simulations with more negative $\Psi_{min}$ (< -200 MPa) and higher Q10 ($\geq$ 5) tended to have lower RMSE in comparison with measured weekly mean NEE (Fig. 4). As changes to $\Psi_{min}$ and Q10 had opposing impacts on the magnitude of simulated fluxes, different pairs of parameter values gave similar results. This counterbalancing effect strongly influences identification of an appropriate parameter space, e.g. simulations using a wide range of $\Psi_{min}$ with lower Q10 more greatly overestimated measured NEE during freeze-up and thaw than simulations with higher Q10 values (Fig. 4). Overestimation of simulated NEE particularly impacted cumulative NEE during freeze-up in 2017-18 (Fig. 5), with a reduction in December to mid-March NEE compensating for freeze-up overestimations; using mid-range values of $\Psi_{min}$ (-20 MPa) produced similar simulated and measured total cumulative non-growing season NEE.

## 4.0: Discussion

### 4.1: NEE variability

The default parameterisation of CLM5.0 prevented simulation of soil respiration for most of the snow-covered non-growing season, leading to negligible simulated NEE, contrary to broadly positive patterns of measured NEE. Application of the Sturm et al. (1997) snow thermal conductivity parameterisation reduced simulated soil temperature biases (Dutch et al., 2022; Royer et al., 2021), which reduced the proportion of the snow-covered non-growing season for which simulated NEE was zero. Other TBMs have shown sensitivity of simulated NEE to snowpack representations, with improvements to the representation of the snowpack (including a multi-layer snowpack with variable, as opposed to prescribed, snow thermal conductivity) in LPJ-GUESS improving the simulation of wintertime NEE (Pongracz et al., 2021).

Cumulative snow-covered non-growing season NEE is not only dependent on parameterisation of snow thermal conductivity, but also the timing of snow onset at the start of the winter. In 2018 – 19, when the snow-on date was 3 weeks earlier than the previous year, soils cooled more slowly due to thermal insulation against cold atmospheric air, leading to greater cumulative NEE. This was particularly evident for simulations using the Sturm thermal conductivity parameterisation, which better represents the early winter formation of low thermal conductivity basal snowpack depth hoar layers. Interannual variability in snow conditions are reflected in simulated fluxes, further substantiating the importance of improving simulations of Arctic snowpacks. Biases and uncertainties in simulated snow mass (e.g. Kim et al., 2021; Mudryk et al., 2020), are likely to influence



soil temperature (Dutch et al., 2022), heterotrophic respiration and $CO_2$ fluxes, particularly on regional scales (Tao et al., 2021). Improving the representation of snow and soil conditions, or at least how these relate to respiration at the start of the snow-covered non-growing season, is also important as this is likely to be the most biologically active part of the season with comparatively high rates of soil respiration (Commane et al., 2017; Olsson et al., 2003).

Simulated NEE increased considerably after the start of snowmelt, regardless of parameter choices, but was less rapid for
simulations with larger negative $\Psi_{min}$. Simulated NEE was most likely too positive at the end of the winter season due to delayed onset of simulated photosynthesis (Birch et al., 2021) and not well matched to trends in measured NEE, which decreased from late April through May. Simulated gross primary productivity was zero for the entirety of the snow-covered period, but the pattern of decreasing measured NEE during thaw suggested that photosynthesis could be occurring before snow had completely melted out, which has been observed at similar Arctic locations (Finderup Nielsen et al., 2019; Larsen et al.,
2007; Starr and Oberbauer, 2003).

## 4.2: Parameterisation of soil moisture, temperature and respiration

Of the three parameters investigated, $\Psi_{min}$ had the largest impact on the simulated snow-covered non-growing season NEE. Without changes to $\Psi_{min}$, simulated soil moisture limited soil respiration, meaning simulated NEE was near-zero for the majority of the snow-covered non-growing season. Accurate simulation of a moisture threshold to soil respiration is important
as moisture acts as a key control on soil respiration (Orchard and Cook, 1983), particularly in the shoulder season before snow onset (Liu et al., 2020) and in frozen soils (Öquist et al., 2009). Consequently, changes to soil moisture content have a strong influence on simulated soil respiration and wider carbon cycling (Chadburn et al., 2017). CLM5.0 represents limitation of respiration in frozen soils by an unavailability of liquid water (Lawrence et al., 2018), as shown by the strong dependence of simulated fluxes on $\Psi_{min}$. However, CLM5.0 has known deficiencies in simulating soil moisture in high-altitude and high-
latitude environments (Deng et al., 2020; 2021; Schädel et al., 2018), overestimating soil moisture when soils are frozen (Deng et al., 2020) and with soil heating leading to increased soil dryness, as opposed to observed increases in soil wetness (Schädel et al., 2018). Soil moisture biases may even have been exacerbated by model development, with Deng et al. (2020) finding a greater difference between simulations and observations for CLM5.0 than CLM4.5.

Even in frozen soils, liquid water can be present within the soil matrix, enabling respiration at temperatures as low as -18 °C
(Elberling and Brandt, 2003), whereas when $\Psi_{min}$ exceeds $\Psi$ simulated soil respiration ceases at warmer temperatures than -18 °C. Recent findings from Liang et al. (2022) suggest that mineral soils should be able to respire below a $\Psi$ of -10 MPa, suggesting a $\Psi_{min}$ below -10 MPa would be more physically representative than the current CLM5.0 default of -2 MPa. Tao et al. (2021) also highlighted the unsuitability of such a high $\Psi_{min}$, the default $\Psi_{min}$ of -10 MPa in E3SM, a five times larger negative $\Psi_{min}$ than the CLM5.0 default, prevented simulation of respiration when soil temperatures were sub-zero and failed
to allow the accurate simulation of wintertime respiration in permafrost tundra environments. A more mechanistic approach, e.g. Yan et al. (2018), where respiration increases linearly from zero as soon as soil moisture is not zero (Chadburn et al., 2022) may produce more appropriate simulations of soil respiration in tundra environments than the commonly used thresholding





approach of CLM5.0. The use of a $\Psi_{min}$ threshold may still be appropriate if decomposition does not automatically drop to zero when the threshold is reached, for example, the $r_w$ scalar in JULES drops to 0.2, not zero, when $\Psi$ is lower than $\Psi_{min}$
(Burke et al., 2017), allowing for wintertime decomposition.

The impact of changing Q10 in CLM5.0 was lower than in other TBMs; smaller changes to Q10 had a larger influence on E3SM simulated fluxes at similar Arctic tundra sites (Tao et al., 2021). At most negative $\Psi_{min}$ values, higher Q10 values were required to simulate soil respiration more accurately, similar to Tao et al. (2021) who found that a Q10 66% larger than the default of 1.5 led to improved simulations of wintertime soil respiration for sites in the Alaskan tundra. As observed Q10
changes with temperature, it may be more appropriate to generate Q10 at each timestep as a function of soil temperature, an approach already undertaken in other TBMs such as CLASSIC (Melton and Arora, 2016; Wu et al., 2016). By using both soil moisture and soil temperature to parameterise Q10, Kim et al. (2019) found an improvement to negative ecosystem respiration biases compared to the use of a Q10 of 1.5 in global CLM4 simulations. However, Byun et al. (2021) states that standard Q10 functions fail when describing the relationship between temperature and $CO_2$ production of frozen soils, and so the use of a
Q10 function may not be the most appropriate way to model the relationship between soil respiration and temperature at sites such as TVC. Alternative parameterisations of $r_T$ (such as RothC; Jenkinson, 1990) may provide a more appropriate description of the relationship between temperature and soil respiration, as has been suggested for other TBMs such as JULES (Burke et al., 2017). This may not lead to improved model performance; Tao et al. (2021) tested non-Q10 parameterisations of the soil temperature-respiration relationship in the CLM-based E3SM and found that a Q10 parameterisation gave the best result for 3
of their 4 Alaskan Arctic sites. Although limited observational data of soil respiration limits the assessment of suitability of Q10 functions (Kim et al., 2019), testing additional parameterisations (as opposed to just Q10) may give insight that could improve the simulation of NEE at Arctic tundra sites in CLM5.0.

## 5.0: Conclusion

The default parameterisation of CLM5.0 did not reproduce the broadly positive measured NEE during snow-covered non-
growing seasons at our Arctic tundra site, despite widely documented midwinter $CO_2$ emission at numerous sites across the Arctic tundra (Natali et al., 2019; Virkkala et al., 2021). An overly conservative moisture threshold limiting soil respiration in frozen soils was the most likely explanation for the lack of simulated soil respiration for the majority of the snow-covered non-growing season. Furthermore, the default parameterisation of CLM5.0 did not capture sub-seasonal patterns of measured NEE. Simulated NEE was too high towards the start of the snow-covered non-growing season, regardless of parameter values tested.
Initial conditions at freeze-up are important in determining the magnitude of cumulative NEE for the entire snow-covered non-growing season, with changes to all parameters tested having the greatest impact at this time as the insulative capacity of the snow has not yet been reached.

Reducing soil temperature biases in CLM5.0 through a change to the parameterisation of snow thermal conductivity, from Jordan (1991) to Sturm et al. (1997), increased the magnitude of simulated NEE during the snow-covered period. However,



without improvement to the minimum soil moisture threshold, other parameter changes had very little impact on simulated NEE. The default $\Psi_{min}$ of –2 MPa was not appropriate for Arctic environments, with a five times larger negative $\Psi_{min}$ producing snow-covered non-growing season NEE more similar to measured NEE. Not only did the default parameterisation of $\Psi_{min}$ prevent wintertime respiration, poorly representing seasonal and annual carbon budgets and dynamics, it may also have longer term implications for the simulation of soil carbon turnover and the state of permafrost, limiting the reliability of longer term

climate simulations. Larger positive Q10 had an opposite impact on simulations than larger negative $\Psi_{min}$, with larger Q10 depressing the magnitude of simulated NEE. Adjustments to both parameters in tandem provided the greatest improvement to simulated NEE, with larger negative $\Psi_{min}$ and larger positive Q10 simulating greater NEE during the snow-covered non-growing season.

**Code & Data Availability**

Code and data to produce figures is available at: https://github.com/V-Dutch/CLMWinterFlux_TVC

**Author Contribution**

Investigation, Formal Analysis, Writing - Original Draft preparation; VRD, NR, LW, OS. Supervision; NR, LW, MS, CD,

RK. Data acquisition; OS, GHG, BW, JB, MD. Data Planning; PM. Software; LW. Funding acquisition; NR, OS, PM. All authors were involved in reviewing and editing prior to submission.

**Competing Interests**

The authors declare no competing interests.


**Acknowledgements**

VRD was funded by an RDF Studentship from Northumbria University and the Northern Water Futures project. The eddy covariance and supporting measurements at Trail Valley Creek were funded through the Canada Foundation for Innovation, the Canada Research Chairs Program, and a Natural Sciences and Engineering Research Council of Canada Discovery Grant

awarded to OS and PM.



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




| *Parameter to Adjust* | *Values* | | | |
|---|---|---|---|---|
| **Q10** (Temperature sensitivity of soil respiration) | 1.5 (*Default*) | 2.5 | 5 | 7.5 |
| **Ψmin** (Moisture threshold for soil respiration) | -2 MPa (*Default*) | -20 MPa | -200 MPa | -2000 MPa |
| **Snow Thermal Conductivity** | Jordan (1991) (*Default*) | Sturm et al. (1997) | | |

**Table 1: Parameters included in sensitivity analysis and the range of values sampled.**

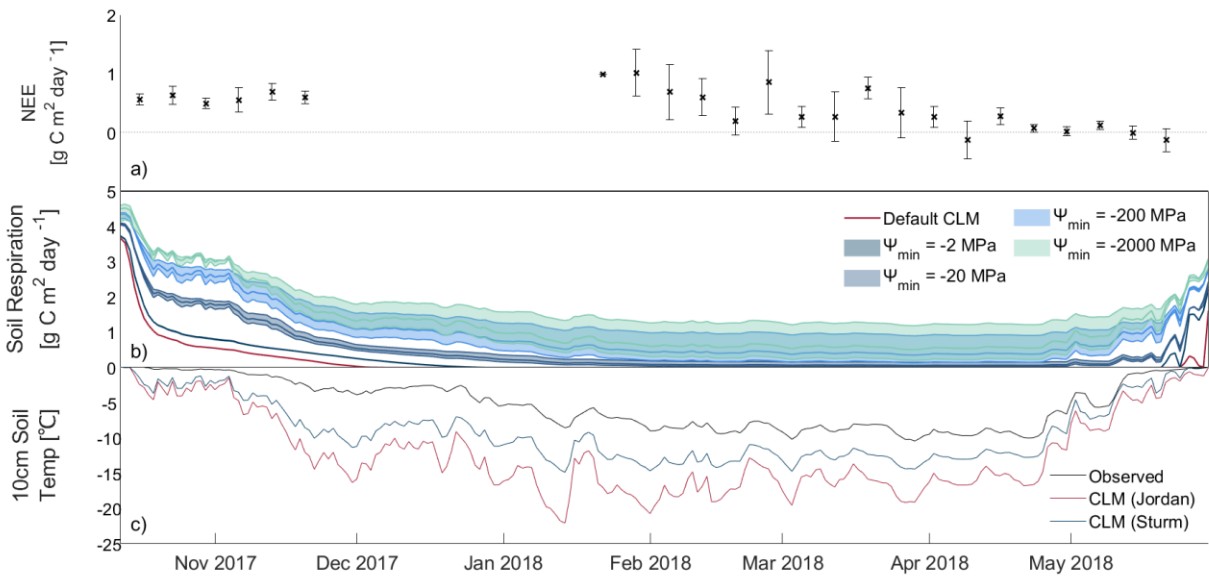

**Figure 1: a) Mean (crosses) and uncertainty (as per Lasslop et al. (2008); error bars) of measured NEE at weekly intervals. b) Simulated soil respiration. The default simulation (red) uses the Jordan (1991) parameterisation of snow thermal conductivity, and blue colours represent simulations using the Sturm et al. (1997) parameterisation of snow thermal conductivity. Darker blue colours represent less negative $\Psi_{min}$ and paler blue colours represent more negative values of $\Psi_{min}$. Shaded areas on b) represent the range of respiration fluxes for simulations using the Sturm et al. (1997)**
**snow thermal conductivity and the same $\Psi_{min}$, but with different values of Q10 (1.5, 2.5, 5.0, 7.5). c) 10 cm soil temperatures, both observed (black) and simulated using both the default Jordan (1991; red) and Sturm et al. (1997; blue) snow thermal conductivity parameterisations.**



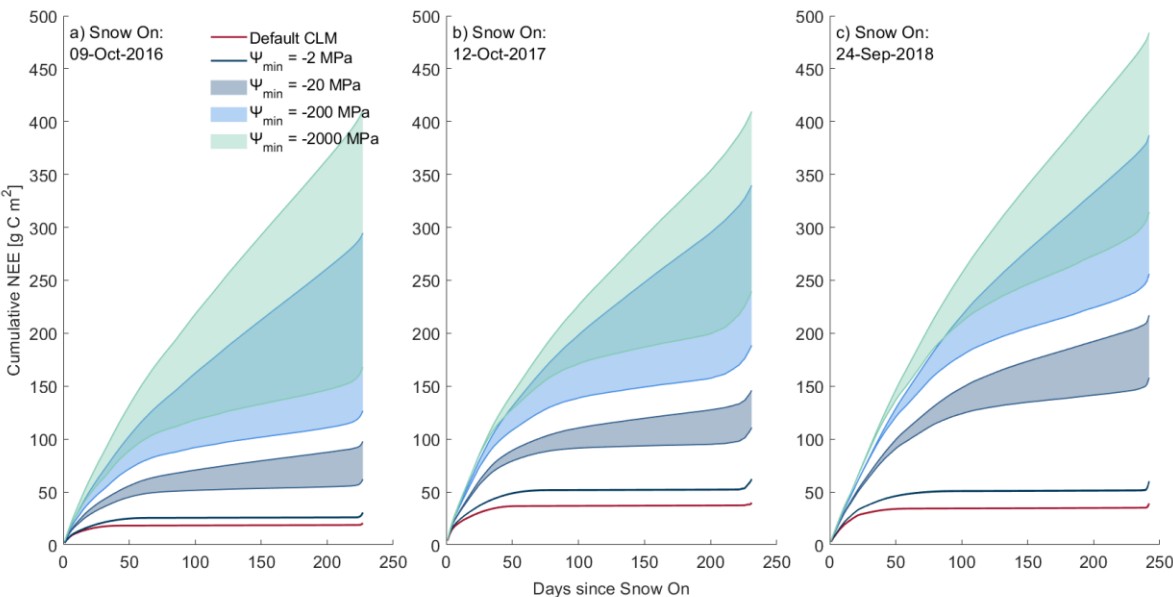

**Figure 2: Cumulative Net Ecosystem Exchange (NEE) for the simulated snow cover duration of a) 2016 – 17 (227 days), b) 2017 – 18 (231 days), and c) 2018 – 19 (242 days) from the ensemble of simulations. Blue colours represent simulations using the snow thermal conductivity parameterisation of Sturm et al. (1997), with darker colours for less negative $\Psi_{min}$. The shaded areas represent the range of Q10 (1.5 – 7.5) for each $\Psi_{min}$. The dark red line represents the default CLM snow thermal conductivity parameterisation of Jordan (1991).**






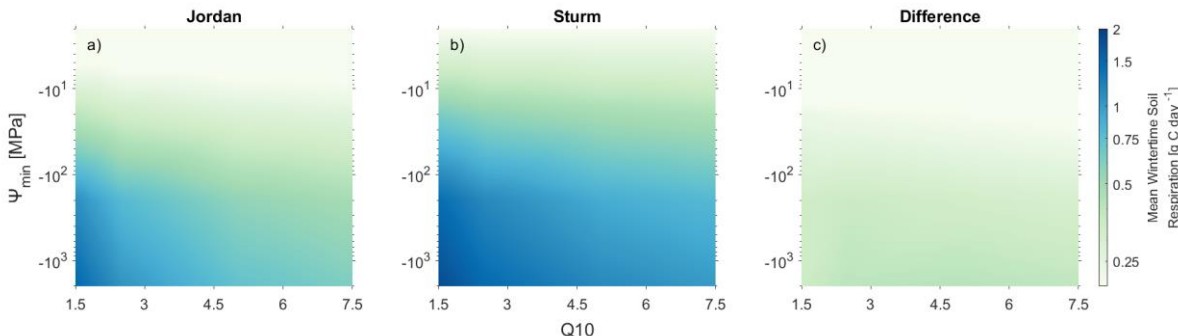

**Figure 3: Contour plots showing the relative influence of $\Psi_{min}$ and Q10 on the simulations of mean soil respiration for all 3 snow-covered non-growing seasons using the snow thermal conductivity parameterisations of a) Jordan (1991) and b) Sturm et al. (1997). The difference between the two snow thermal conductivity parameterisations is shown in c).**




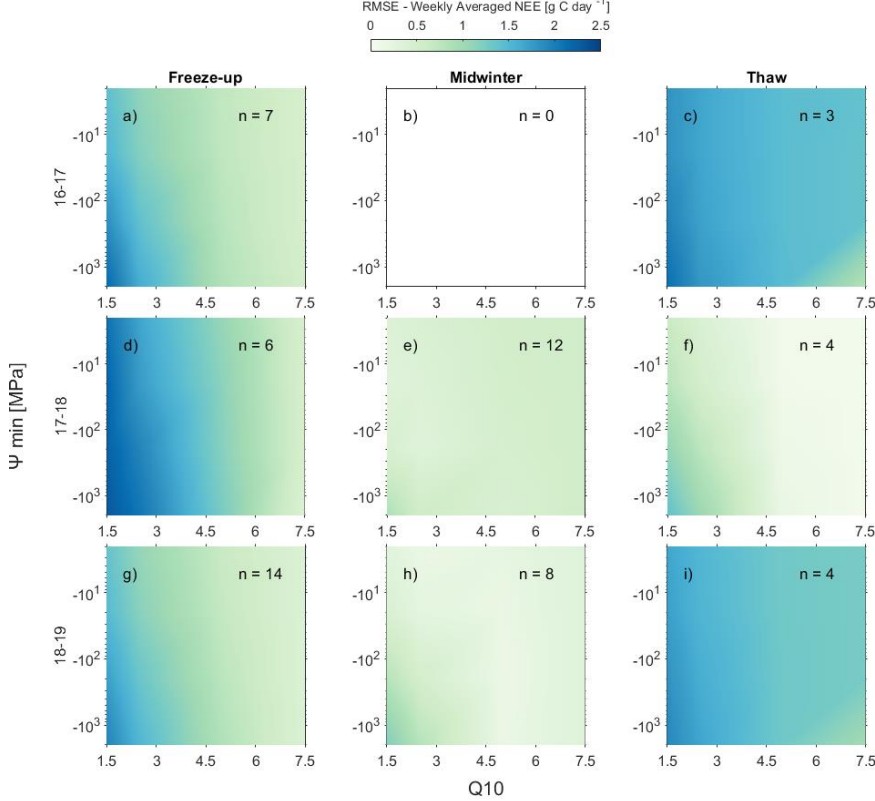

**Figure 4: Evaluation of the impact of $\Psi_{min}$ and Q10 parameterisations on simulated Net Ecosystem Exchange (NEE) during freeze-up (a, d, g), midwinter (b, e, h) and thaw (c, f, i) periods of each snow-covered season for simulations using the (Sturm et al., 1997) snow thermal conductivity parameterisation. The number of weekly averages included in each panel are denoted by n values.**




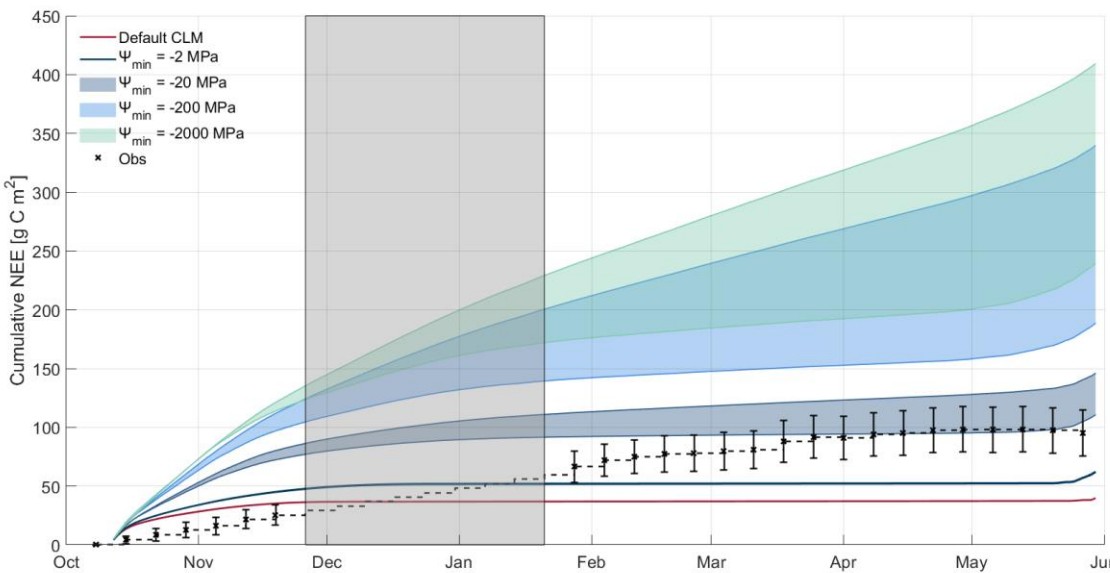

**Figure 5: Cumulative Net Ecosystem Exchange (NEE) for winter 2017 – 18. The black crosses show cumulative weekly measured NEE, with error bars representing measurement uncertainty as per Lasslop et al. (2008). The grey area from late November to late January denotes the period where no NEE observations are available. Across this section, an average value for the six weeks before and after the gap is used to estimate cumulative NEE. Curves show the simulated cumulative NEE, with blue colours representing simulations using the snow thermal conductivity parameterisation of Sturm et al. (1997), with darker colours for less negative $\Psi_{min}$. The shaded areas for these curves represent the range of Q10 (1.5 – 7.5) for each $\Psi_{min}$. The dark red line represents the default CLM snow thermal conductivity parameterisation of Jordan (1991).**