# Peer review of "Simulating net ecosystem exchange under seasonal snow cover at an Arctic tundra site"

_EGUsphere, 2023_

## Author Comment (AC1)

**Author Response for *"Simulating net ecosystem exchange under seasonal snow cover at an Arctic tundra site"*, Victoria R. Dutch et al.**

We would like to thank the editor and the reviewers for taking the time to read and comment on the original manuscript. We provide a response to these comments below. For ease, comments are in black and our responses in purple. This is an intermediate response, we have undertaken some additional simulations based on the comments of the first review and have another one still to come, which may result in additional changes to the manuscript beyond those outlined here. Sections which are likely to change further are clearly noted and things still to add are highlighted.

**Reviewer 1:**

Dutch and co-authors look into simulations of winter time soil temperatures and heterotrophic respiration (HR) fluxes that are simulated at a EC flux site in NWT, Canada with the Community Land Model, version 5. Observations suggest positive NEE fluxes throughout much of the very cold winter, but the model simulates cold soil biases and zero HR fluxes. Building off their work looking at snow thermal properties, they explore alternative parameterizations for snow thermal conductivity, minimum soil water potential of soil organic matter decomposition, and higher temperature sensitivity (q10) of soil organic matter decomposition to capture positive NEE fluxes with the model. The paper is well written and clear.

**Major concerns:**

I'm concerned, however, that the experimental approach seems like it is somewhat putting the cart (NEE & snow thermal properties) before the horse (simulated soil temperatures). Specifically, it seems that potential biases in soil temperature really need to be addressed before trying to adjust the sensitivity of heterotrophic respiration fluxes.

Line 187, this strikes me as a potential weakness if soil properties from the global surface datasets are not consistent with real soils at the site. It's OK to use these data if no observational data are available, but calibrating snow thermal properties to generate warmer soils seems odd if the soil thermal and hydraulic properties are not being estimated correctly in the first place because of poor calibration to local soil properties. Specifically, I wonder if higher organic matter concentrations in surface soils would prevent the cold biases you're seeing, especially early in the simulation?

No modelling approach currently simulates soil, snow and $CO_2$ well in the snow-covered non-growing season. Permafrost models often have a crude representation of snow and lack a representation of $CO_2$, snow models do not include $CO_2$ and often have a crude representation of soils. It therefore makes sense to approach these one at a time. Snow is a major control on soil temperatures over seasonal cycles, and soil temperature simulations

have been considerably improved by the adjustment to snow thermal properties. That said, as a result of these review comments we have run additional simulations to investigate the impact of changes to soil properties to better match observational data. Ongoing work elsewhere by our collaborators seems to suggest that changes to the soil organic matter will have a lower impact on simulated soil temperatures than the impact of snow throughout the course of the winter. Site measurements (and thus the new simulations) give lower soil organic matter contents than seen by default in CLM, and so adding additional SOM to compensate for the soil temperature bias does not seem advisable.

On the next page, we show the results of 6 new simulations, using a combination of 3 different SOM profiles (SOM1 = CLM Default, SOM2 = Cryogrid default, SOM3 = Observed SOM interpolated onto CLM soil layers; Boike et al. [Unpublished]) and 2 different soil texture profiles (SC1 = CLM Default, SC2 = Observations interpolated to CLM soil layers). These simulations all use the sturm snow thermal conductivity parameterisation, a $\Psi$min value of -20 and a Q10 value of 2.5. All six simulations use the same spin-up method as the simulations in the paper.

In these new simulations (shown below), changes to soil organic matter have a limited impact on both soil temperature and soil respiration within the non-growing season. Changes to the soil texture (black vs red) appear to have a greater impact on simulated soil temperature than changes to the soil organic profile (different line styles). The majority of this variability between the different soil characteristics is restricted to the early snow season, highlighting the importance of the zero curtain period as a control on total seasonal NEE. Different amounts of soil organic matter lead to changes in NEE as a result of changes in the insulative properties of the soil as opposed to any change to the supply of labile carbon for respiration.

| | Modelled - observed 10 cm soil temperature (October 2017 - May 2018) | | | |
|---|---|---|---|---|
| Soil Profile | Mean | Min | Max | Std. Dev. |
| SC1 - SOM1 | -3.3 | -7.9 | 7.1 | 1.9 |
| SC1 - SOM2 | -4 | -9.3 | 9.2 | 2.3 |
| SC1 - SOM3 | -3.5 | -8.6 | 10 | 2 |
| SC2 - SOM1 | -2.4 | -5.3 | 6.9 | 1.3 |
| SC2 - SOM2 | -3.5 | -8.1 | 8.1 | 1.9 |
| SC2 - SOM3 | -2.8 | -7.4 | 8.9 | 1.7 |

We note that simulated snow depths are underestimated in the winter of 2017-18, likely due to gauge undercatch (Dutch et al., 2022), contributing to the cold mean soil temperature bias seen in all soil profile configurations.

[Figure]

[Figure]

| nlevsoi | interface: upper (m) | interface :lower (m) | SOM1 ORGANIC (kg/m3) | SOM2 ORGANIC (kg/m3) | SOM3 ORGANIC (kg/m3) | SC1 SAND % | CLAY % | SC2 SAND % | CLAY % |
|---|---|---|---|---|---|---|---|---|---|
| 1 | 0 | 0.02 | 130 | 77.94 | 33.5 | 28 | 36 | 14 | 17 |
| 2 | 0.02 | 0.06 | 88.7 | 65.67 | 33.5 | 28 | 36 | 14 | 17 |
| 3 | 0.06 | 0.12 | 56.5 | 64.82 | 33.5 | 28 | 36 | 20 | 19 |
| 4 | 0.12 | 0.2 | 35.4 | 63.29 | 33.5 | 28 | 36 | 25 | 12 |
| 5 | 0.2 | 0.32 | 22.1 | 55.32 | 22.1 | 28 | 36 | 25 | 16 |
| 6 | 0.32 | 0.48 | 13.8 | 46.62 | 13.8 | 28 | 36 | 25 | 16 |
| 7 | 0.48 | 0.68 | 8.6 | 38.98 | 8.6 | 28 | 36 | 25 | 16 |
| 8 | 0.68 | 0.92 | 5.4 | 29.87 | 5.4 | 28 | 36 | 25 | 16 |
| 9 | 0.92 | 1.2 | 0 | 0 | 0 | 28 | 36 | 25 | 16 |
| 10 | 1.2 | 1.52 | 0 | 0 | 0 | 28 | 36 | 25 | 16 |

However, none of these potential sources of uncertainty lead to a variability in soil respiration or NEE greater than the uncertainty estimates of the eddy covariance observations. The soil profile we have used in the publication (solid black line) sits within the middle of the range of profiles tested, whereas the in-situ observations may sit closer to the red line (improved soil texture) in the snow covered non-growing season of 2017-2018. Soil conditions at Arctic tundra sites are highly heterogeneous; the level of variability between these simulations can be seen as what uncertainties in deriving grid square average value introduce to CLM.

We will add an additional section to the discussion discussing the impact of the soil properties, including the summary table on page 2. The Supplement will be revised to include the table describing soil profiles (shown on page 3), and the graph showing the variability in soil respiration throughout the snow covered non-growing season as a result of changes to the soil profile.

Soils still seem to be getting pretty cold, even after modifying snow thermal conductivity parameterisations (Fig 1c). Subsequent modifications to moisture and temperature sensitivity of HR, therefore seem to be compensating for persistent soil temperature biases that ultimately make the cumulative wintertime NEE fluxes worse (shaded regions of Fig 5). Indeed, the authors settle in on really high q10 and psi min values to get any soil respiration out of the model but still have 5C temperature biases during the time of year when they're trying to get respiration fluxes out of really cold soils.

We will revisit our conclusion to more explicitly state the physical implausiblity of the highest Ψ min parameter values used. Whilst we do not settle on or recommend any particular combination of parameter values, given the short duration (three winters) and single site nature of this study, some of the tested values are less reasonable than others.

I don't really know what to suggest for this critique. Ideally, additional simulations could be run that either modify organic profiles on the surface dataset or explore alternative pedotransfer functions to calculate soil thermal properties. If additional simulations are not feasible, some additional discussion on addressing soil temperature biases seems warranted.

See above. Additional simulations with modified organic profiles have been run but making changes to the soil thermal properties within CLM through the use of different pedotransfer functions, is beyond the scope of this paper.

I wonder if the simulations are really spun up correctly? Specifically cumulative wintertime NEE is very high with changes to minimum soil water potential (Line 237; Fig 2). If the system is really at steady state this implies that there's much stronger photosynthetic uptake in the growing season for these simulations. Is this true?

The spinup protocol that we have undertaken reduces the variability in key soil and carbon parameters (NEE:NPP Ratio, Soil Carbon Pool, Total Ecosystem Carbon, SOM Carbon) to a tolerable range. When preparing for this paper, we also undertook a test run with a spin up period of 1012 years (as opposed to the 512 years used herein) and found no discernible difference between the stability of the carbon pools between the two spin up lengths and therefore used the shorter spin up to improve computational efficiency. We also note that the length of the model spin up is greater than that used for many other CLM papers in similar environments, such as that of Birch et al., 2021, where GPP is shown to stabilise within 20 years.

However, we have run an additional simulation using the accelerated decomposition scheme, and show a comparison of this with an otherwise equivalent simulation from our paper below. Changes as a result of the change in spin up protocol are of a similar magnitude to those resulting from a change in soil profile (shown on page 3), and also within the uncertainty of the eddy covariance observations. The use of the accelerated decomposition scheme leads to a lower amount of soil carbon storage, but this still only leads to differences in NEE that are within observational uncertainty for the majority of the winter. This results in a reduction of the cumulative total NEE for the most extreme case ($\Psi$min = -2000) by less than 20%. Small differences in soil temperature also occur as a result of the change in spin up, these are likely a result of numerical instabilities. We will also carry out one more simulation using $\Psi$min and Q10 values that are likely to be more physically representative ($\Psi$min = -20, Q10 = 2.5) but as this simulation shown here is for the most extreme case, we do not expect the impact of the use of the accelerated decomposition scheme to be as great for this upcoming simulation.

[Figure]

Additionally, cumulative NEE shown in Figure 2 is only cumulated from the snow-on date. No growing season values are shown or included anywhere within the paper.

**Minor and technical comments:**

Lines 53-84.  This is a great literature review of some of the things that could be wrong with high latitude C fluxes that are simulated by land models, but as written it kind of gives the reader whiplash.  It reads like a list of shortcomings that's scattered around multiple processes (soil temperature, phenology and plant productivity, plus temperature and moisture sensitivity of heterotrophic respiration).  Maybe breaking this up into a few paragraphs with clear topic sentences to guide the reader?  I wish I had more concrete suggestions but would encourage revisions to make this text more reader friendly.

I've added a few line breaks to split this into 3 sections; the first on soil temperature biases caused by poor simulation of snow, one on the empirical relationships between soil temp/moisture and respiration and a final one on plants and seasonal carryover.

Line 64, these are both CLM references, but not "other Earth System Models"?

These two papers refer to different versions of CLM (CLM5.0 and CLM versions 4, 4.5 and 5.0), whereas the final reference in the sentence refers to 4 models including CLM4.5. The sentence has now been changed to read: " … has been previously noted in CLM (Birch et al 2021; Wieder et al., 2019) and other earth system models to due model underestimates of $CO_2$ uptake by Arctic vegetation (Rogers et al., 2017).

Line 114, what is the forcing height, ZBOT, that's used for these single point simulations?

2 m. This will be added to the supplementary information.

Line  136, was USTAR filtering done on the observations (e.g. https://ameriflux.lbl.gov/data/flux-data-products/data-qaqc/ustar-filtering-module/)?

Yes, U* filtering was done using a threshold of 0.1ms^-1 (also given on line 22 of the supplementary information).

Line 209, was the accelerated decomposition (AD) mode of the model used for this period? If so for how long? How long did you run in postAD mode?  See Lawrence et al. 2019

Accelerated decomposition was not used to spin up these simulations. To spin up, the model was run for 512 years using cyclical meteorological data. We have tested the impact of using the accelerated decomposition mode, as shown above and find that differences in the non-growing season are within observational uncertainty.

Line 211, this is worded oddly,  Lawrence et al. recommend changes in ecosystem C < 1gC/m2/y for steady state conditions.

Please see comments above about the model spin up. The phrasing of this will be revisited.

Line 218, supplementary figures should be numbered in the order they are introduced in the text.

Section 1 of the supplementary material (referenced in line 134) includes and is meant to implicitly refer to Supplementary Figures 1 & 2. The inclusion of these figures when referring to this section is now explicitly mentioned in line 134.

I find Figure 1 really hard to read. Can lines be thicker, and maybe the magnitude of the flux axes (NEE and HR) be standardized to the same scale?

[Figure]

The vertical scale on plots 1a and 1b has been standardised and the thickness of the lines on plots 1c and 1d has been increased. As mentioned in your next point, a snow depth plot has also been added (new 1c).

I kept wondering about the evolution of snowpack that is simulated by CLM at the site. Can this be shown in Fig 1? (see also line 222 & 290). Specifically, I'd assume that snow is pretty shallow in the late-fall / early winter when surface soils are experiencing pretty cold biases (Oct-Dec, Fig 1c). Accordingly, changes to snow thermal properties has pretty minimal effects on soil temperature.

Snow at TVC is reasonably shallow (typically under 50cm) for the entire snow-covered season. However, early season snow cover is very important in insulating soil temperatures and in initiating soil temperature and moisture conditions for the rest of the winter. The early snow season is when strong temperature gradients lead to the initial formation of the highly

insulative depth hoar layer, thereby increasing the importance to simulations of soil temperatures.

How do soil temperature profiles evolve in the simulations?  Given cold surface temperature biases I'm assuming that deeper layers get quite cold too, but in reality does it take longer for these deeper horizons to freeze?  Could these deeper horizons be the source of positive wintertime NEE fluxes at the site?  Would this also be true in the model, or is CO2 produced deeper in CLM not allowed to move through frozen soil layers?  See also Knowles et al. 2019.

Simulations of deep soil temperatures are not shown in the paper as we only have observations down to a depth of 20cm for comparison. Profiles of soil temperatures for winter 2017-18 can be seen in the figure below, with warmer temperatures indeed found in deeper soil horizons. Deeper soil horizons do freeze more slowly, with soil below the active layer (~1 m) permanently frozen.  $CO_2$ emissions from deeper soil layers may be physically plausible, but soil respiration from CLM is not output by individual layers so precisely where this is simulated in the soil column is unclear. Soil water potentials in deep soil layers fall below the value of $\Psi$min later in the winter than those at the surface, so deeper soil respiration would make sense. An acknowledgement of this will be added to the discussion.

[Figure]

To the point above, I'm surprised to see positive NEE fluxes in Feb-April when observed surface soils are so cold.  I'd be surprised if microsites in surface soils at -10C could sustain such high fluxes.  Can the authors speak to the physical and biological processes that are likely at play here?

Production of $CO_2$ by soil microbes has been observed at temperatures lower than -40$^{\circ}$C in laboratory environments (Panikov et al., 2006) and down to approximately -20$^{\circ}$C in field

studies (Natali et al., 2019). Films of liquid water can exist around soil particles and in soil pore spaces (Hayashi et al., 2013), sustaining microbial activity well below 0$^\circ$C (Henry, 2007; Elberling & Brandt, 2003). However, production and release of $CO_2$ may become decoupled in frozen soils, with soil cracking with the deeper penetration of the freezing front potentially releasing stored $CO_2$ produced when levels of respiration are higher under comparatively warmer conditions. Mention of these processes in the discussion will be expanded upon.

Additionally, during March 2022, we measured subnivean $CO_2$ concentrations greater than atmospheric across this site down to soil temperatures of around - 10$^\circ$C using multiple different techniques (Mavrovic et al. [in review] (also found the same for 2021 measurements); Dutch et al. [in prep]).

LIne 245, I'm assuming this claim is in reference to 1b and the difference between when the thin red and thin blue lines hit the 0 line for HR fluxes?   It's really hard to see, but to my eye this looks like ~1 month difference.  Regardless, supporting the statement with a reference to the display item seems important.

Yes, this is in reference to Fig. 1b. This has been updated, with dates of zero soil respiration and a reference to this figure added to this sentence.

I like using display items to support claims made in the text, but none are referenced in the discussion.  This is more of a stylistic comment, but it helps focus readers on highlights of the findings being presented in this study. Line 350, this is one example of an interesting claim that could be supported by results presented in the paper?

We appreciate your thoughts, but it is within our stylistic remit to not do this. Figures are cross-referenced elsewhere in the paper, but the discussion section is where we link our results to works of others.

**Reviewer 2:**

The manuscript describes a study in which land surface model CLM5.0 was used to simulate NEE of an Arctic site, from which eddy covariance flux measurements and meteorological data were available. The default parameterisation of the model was known to produce too low wintertime NEE in Arctic environments, and the authors tested how sensitive the soil respiration simulation is to certain parameters. They found that a different parameterisation of snow thermal conductivity and lowering the minimum soil moisture required to simulate soil decomposition improved the NEE simulation most. The simulation was less sensitive to changes in Q10.

This is a clearly written manuscript that communicates nicely the results of the parameter test. Also the topic, how to improve wintertime greenhouse gas simulations in land surface models, is important. However, I think the paper content is a bit too plain and some additions are necessary. Also, I find some conclusions could be justified more. Below are my suggestions and comments.

A general comment:

You did the parameter sensitivity test at one site, Trail Valley Creek. I think the value of the study would increase a lot if you could add some Arctic site(s) and show that the new parameterisation improves wintertime NEE simulation also there, compared to the default parameters. Can you find some additional flux and met data sets from the Arctic to run your model with, e.g. via the paper you refer to: Virkkala et al. 2021?

Having this combination of detailed measurements of snow and soil alongside wintertime eddy covariance measurements of $CO_2$ flux is incredibly rare - we are unaware of any other sites with such detailed measurements of all 3 types publicly available. Of the list of Arctic flux towers compiled by Pallandt et al. (2021), only 20% remain operational throughout the winter, and of these none are known to have both complete meteorological records, as well as comprehensive and coincident snow and soil measurements. Virkkala et al. statistically upscales $CO_2$ flux across the Arctic Boreal zone due to this lack of measurements.

We are aware of the spatial and temporal limitations presented by our study. Limited data availability prevents meaningful comparison on a multisite scale, but because such data are so hard to come by, this study still provides valuable insight despite its limited extent. Work is ongoing in comparing the application of this change in snow thermal conductivity to distributed soil temperature measurements, but this is a separate effort and the evaluation data to also compare to $CO_2$ fluxes on these larger spatial scales is not currently available.

Detailed comments:

line 22: You mention snow properties here, but in the Methods section the only snow property that you list is general snow depth (reported in King et al. 2018). Were there more? Was this information needed for the snow thermal conductivity parameterisation?

Yes, additional snow properties measured included snow density, penetration force and direct measurements of snow thermal conductivity. These measurements are compared to each other and to CLM simulations in our previous work, where we decide the substitution of the default CLM snow thermal conductivity parameterisation with that of Sturm is more appropriate for this environment. Snow thermal conductivity is generally parameterised as a function of snow density; both the default (Jordan) and Sturm parameterisations are different empirical relationships between snow density and snow thermal conductivity.

line 41: Please specify what $CO_2$ fluxes. Non-growing season fluxes or total?

Edited to specify this refers to winter (i.e. non-growing season) fluxes.

line 63-64: How well has CLM5, with the current parameterisation, simulated wintertime NEE in other than Arctic environments? E.g. boreal?

Studies of the performance of CLM5 exclusively within Arctic environments or during the wintertime are  limited. Birch et al. (2021) focuses on the performance of CLM5 in the Arctic during the growing season; there is a greater knowledge and availability of evaluation data at this time. Known issues occur with the year-round simulation of NEE in Siberia due to a cold soil temperature bias leading to issues sustaining plant growth in this region (see: https://github.com/NCAR/LMWG_dev/discussions/3).

line 173: This apparently is because frozen water = ice is not part of soil moisture? Please explain it here.

Within the context of soil decomposition, CLM defines soil moisture as soil liquid water content.

line 185: Please state briefly, what the impact of the standard deviation of elevation is in this model experiment.

Changes to the standard deviation of elevation impact the duration for which the entire grid cell is snow covered (as represented in CLM through the use of the snow covered fraction). Without changes to the standard deviation of elevation, small parts of the grid cell do not become snow covered until very late in the season. This has been added to line 194.

line 193-195: When you describe the experiment setup (2.3) you present no details about the parameterisation of snow thermal conductivity. You refer to Jordan (1991) and Sturm et al. (1997), and the reader can of course find the papers and check the details of the

conductivity equation and the parameter values there, but in my opinion, it would be necessary to show them here. They are in focus, and you found that they impact the results, so you should present them also here.

The CLM snow scheme (and the impact of the change in snow thermal conductivity parameterisation) is described in detail in our previous paper (Dutch et al., 2022). Snow thermal conductivity in CLM is described from lines 184 - 189:

"Parameterisation of effective snow thermal conductivity ($K_{eff}$) in CLM5.0 is after Jordan (1991). A quadratic equation is used to infer the relationship between the density of the snow (calculated from the masses of ice and interstitial air) and the thermal conductivity of the snowpack. Other parameterisations of this relationship, typically using different constants in the same quadratic equation (Sturm et al., 1997; Calonne et al., 2019; Yen, 1981), was expanded upon for CLM5.0 in Dutch et al. (2022). These different constants have been calculated from snow samples from different environments, with the parameterisation of Sturm et al. (1997) derived from snowpacks in the Alaskan Arctic."

line 204: The lowest Ψmin, -2000 MPa, is very low indeed. Does your model ever reach that low values, would the effect be the same if you just removed the Ψmin condition completely in the Arctic?

Simulated soil moisture potentials never reach a Ψ value of -2000 within the duration of our experiment.  We note that such a large negative value is physically implausible and will add greater emphasis to this in the final version of the paper. To remove the Ψmin condition would prevent the same relationship between soil moisture and soil respiration from being used - such a change should be functionally the same as the use of a very low Ψmin value but may lead to slightly different results than we see here due to mathematical artefacts caused by changing the relationship between Ψ and soil decomposition from linear to asymptotic.

line 244-246: Apparently because of higher soil temperature and therefore no/less ice? Please explain it here.

Changes to the snow thermal conductivity parameterisation result in warmer simulated soil temperatures. A greater amount of liquid soil moisture is likely available when soils are warmer, but changes impacting the relationship between soil moisture and soil respiration are not discussed in this paragraph.

line 249-250: There's a clear difference between the measured and modelled soil temperature also with the Sturm parameterisation. Please mention it here.

The following sentence is added, with reference to the new snow depth panel on Figure 1 to explain the discrepancy:

Although the cold soil temperature bias is reduced by two thirds through the use of the Sturm parameterisation, we note that soil temperatures still remain lower than measured due to model underestimation of snow depth for the winter of 2017-18 (Fig 1c.)

line 268-273 and 295-296: If I've understood correctly, you suspect that the main reason why your model overestimated NEE in early spring (and early autumn too?) was biased modelling of photosynthesis in the shoulder season? So, what is your conclusion about the greater overestimation of NEE with low Q10: is the low Q10 a worse choice although you know the overestimation also happens because of negligible/no C uptake in photosynthesis?

We suspect that biases in photosynthesis are a contributing factor, but we would not go as far as to say that this is the main reason for overestimated NEE in the shoulder seasons. Lower Q10 values have a larger effect in the shoulder seasons as the rate of change in soil respiration with temperature is lessened - i.e. respiration decreases less as soil temperatures decrease. We do not recommend the default (lowest) value of Q10, rather that larger negative values of Ψmin and larger positive values of Q10 (in comparison to default parameter values) are required to simulate wintertime NEE more adequately.

e.g. around line 287 (or where you think it fits):  Please add some more detailed discussion explaining why and how the Sturm thermal conductivity parameter values affected the results as they did.

Sentence extended to read: This was particularly evident for simulations using the Sturm thermal conductivity parameterisation, which better represents the early winter formation of low thermal conductivity basal snowpack depth hoar layers, providing greater insulation and thus warmer soil temperatures than the default snow thermal conductivity parameterisation used by CLM.

line 295: Why was it less rapid with larger negative Ψmin?

Changes in NEE with the onset of snowmelt (and the subsequent increase in soil moisture) were less rapid for simulations with a larger negative value of Ψmin as the decomposition in these simulations was never soil moisture limited.

line 297-300 and Fig. 1: Can you partition the measured NEE and show here also the measurement-based estimate of soil respiration, not only total NEE? It would be interesting to see how well it fits with the simulation.

Measured fluxes for the majority of the snow covered non-growing season can be entirely attributed to soil respiration. As uncertainties cross the zero threshold in the later part of May, suggesting that there could be photosynthesis at this time, and only at this time. The best way to partition NEE into GPP and ecosystem respiration for northern latitudes involves the use of a bulk partitioning function using two empirical models - one of which is

based on Q10 (Runkle et al., 2013). Comparing CLM`s GPP (which is also Q10 driven) to partitioned fluxes would not only be comparing two model outputs to each other, but two models that use the empirical Q10 equations to each other and would therefore not be appropriate.

line 309-310 and 346: I am slightly confused by your conclusion about modelled soil moisture and the threshold. You say CLM5.0 overestimates soil moisture when soils are frozen, but on the other hand, you recommend lowering the $\Psi$min because with the original $\Psi$min (that you later describe as "overly conservative") soil decomposition ceased too easily. Please clarify this. Did the model overestimate soil moisture also in your simulation?

If CLM overestimates soil moisture and yet still assumes no soil decomposition can occur because soil moisture is too low, the presumed threshold where low soil moisture prevents decomposition must be too high.

line 314-316: I don't understand the sentence. Can you please clarify what do you mean by "when $\Psi$min exceeds $\Psi$", how is it connected to the threshold -18°C?

The soil moisture threshold $\Psi$min is a model parameter, whereas the -18℃ limit is an observational constraint - these two values are not meant to be read as being related to each other beyond the general idea that it should be reasonable to expect that some liquid soil moisture would be available for soil decomposition.

line 317-320: Also here I don't understand this sentence. Can you please check if it is ok.

This sentence will be rephrased.

line 327-329: Please explain why this was. Why did you need higher Q10 with small $\Psi$min?

Although changes caused by a change in Q10 are lower than those caused by a change in $\Psi$min, changes in Q10 value still affect the simulation with low Q10 leading to model overestimation of soil respiration when $\Psi$min is high. As $\Psi$min is a hard threshold, changes to Q10 have no impact if $\Psi$min is still too low for soil decomposition (and subsequent respiration) to occur. Provided $\Psi$ is greater than $\Psi$min, other, untested, combinations of Q10 and $\Psi$min may also produce reasonable simulated NEE.

**References:**

Dutch, V. R., Rutter, N., Wake, L., Sandells, M., Derksen, C., Walker, B., Hould Gosselin, G., Sonnentag, O., Essery, R., Kelly, R., Marsh, P., King, J., and Boike, J.: Impact of measured and simulated tundra snowpack properties on heat transfer, The Cryosphere, 16, 4201–4222, https://doi.org/10.5194/tc-16-4201-2022, 2022

Elberling, B. and Brandt, K. K.: Uncoupling of microbial $CO_2$ production and release in frozen soil and its implications for 460 field studies of arctic C cycling, Soil Biology and Biochemistry, 35, 263-272, https://doi.org/10.1016/s0038-0717(02)00258-4, 2003.

Hayashi, M.: The Cold Vadose Zone: Hydrological and Ecological Significance of Frozen-Soil Processes, Vadose Zone Journal, https://doi.org/10.2136/vzj2013.03.0064, 2013

Henry, H.A.L.: Soil freeze–thaw cycle experiments: Trends, methodological weaknesses and suggested improvements, Soil Biology and Biochemistry, 39, 5, 977-986, https://doi.org/10.1016/j.soilbio.2006.11.017, 2007

Mavrovic, A., Sonnentag, O., Lemmetyinen, J., Voigt, C., Rutter, N., Mann, P., Sylvain, J.-D., and Roy, A.: Environmental controls of non-growing season carbon dioxide fluxes in boreal and tundra environments, Biogeosciences Discuss. [preprint], https://doi.org/10.5194/bg-2023-92, in review, 2023.

Natali, S.M., Watts, J.D., Rogers, B.M. et al.: Large loss of $CO_2$ in winter observed across the northern permafrost region, Nature Climate Change, 9, 852–857, https://doi.org/10.1038/s41558-019-0592-8, 2019

Runkle, B.R.K, Sachs, T., Wille, C., Pfeiffer, E.-M., and Kutzbach, L.: Bulk partitioning the growing season net ecosystem exchange of $CO_2$ in Siberian tundra reveals the seasonality of its carbon sequestration strength, Biogeosciences, 10, 1337-1349, https://doi.org/10.5194/bg-10-1337-2013, 2013

Panikov, N.S., Flanagan, P.W., Oechel, W.C., Mastepanov, M.A. and Christensen, T.R.: Microbial activity in soils frozen to below -39 ℃, Soil Biology and Biochemistry, 38, 4, 785-794, https://doi.org/10.1016/j.soilbio.2005.07.004, 2006